# Inhibition of Malate Dehydrogenase-2 Protects Renal Tubular Epithelial Cells from Anoxia-Reoxygenation-Induced Death or Senescence

**DOI:** 10.3390/biom12101415

**Published:** 2022-10-03

**Authors:** Theodoros Eleftheriadis, Georgios Pissas, Spyridon Golfinopoulos, Maria Efthymiadi, Vassilios Liakopoulos, Ioannis Stefanidis

**Affiliations:** Department of Nephrology, Faculty of Medicine, University of Thessaly, 41110 Larissa, Greece

**Keywords:** ischemia-reperfusion injury, apoptosis, ferroptosis, senescence, Krebs cycle, malate dehydrogenase, acute kidney injury

## Abstract

Ischemia-reperfusion injury is the leading cause of acute kidney injury. Reactive oxygen species (ROS) production causes cell death or senescence. In cultures of primary human renal tubular epithelial cells (RPTECs) subjected to anoxia-reoxygenation, inhibition of the Krebs cycle at the level of malate dehydrogenase-2 (MDH-2) decreases hypoxia-inducible factor-1α and oxidative stress and protects from apoptotic or ferroptotic cell death. Inhibition of MDH-2 decreased reoxygenation-induced upregulation of p53 and p21, restored the levels of the proliferation marker Ki-67, and prevented the upregulation of the senescence marker beta-galactosidase and interleukin-1β production. MDH-2 inhibition reduced the reoxygenation-induced upregulation of ATP, but the alterations of critical cell metabolism enzymes allowed enough ATP production to prevent cell energy collapse. Thus, inhibition of the Krebs cycle at the level of MDH-2 protects RPTECs from anoxia-reoxygenation-induced death or senescence. MDH-2 may be a promising pharmaceutical target against ischemia-reperfusion injury.

## 1. Introduction

Ischemia-reperfusion (I-R) injury is implicated in many human pathologies, from thrombosis-induced cardiac infarction and stroke to low effective blood volume-induced multiorgan failure [1,2,3]. I-R injury is also the most typical cause of acute kidney injury (AKI), with the kidney and especially the renal tubular epithelial cells being quite vulnerable to I-R injury due to their high metabolic demands [4].

I-R injury consists of two consecutive but pathophysiologically distinct phases. During the ischemic phase, the lack of oxygen ceases the electron transport chain (ETC) in the mitochondria lowering the energy supply to the cells. In the following reperfusion phase, the restoration of oxygen supply leads to a burst of reactive oxygen species (ROS) production and cell injury [5]. Among the sources of ROS production, mitochondria play a significant role, and there are two theories about ROS production by the mitochondria under I-R conditions [6]. One of the theories supports that the ROS burst results from superoxide overproduction due to reverse electron transport at the level ECT complex I during reperfusion [7]. The other theory supports that superoxide overproduction results through the leaking of electrons by ECT to oxygen because of the accumulated during the ischemic phase of the electron donors reduced nicotinamide adenine dinucleotide (NADH) and reduced flavin adenine dinucleotide (FADH_2_). Under ischemia, ECT ceases due to the lack of oxygen, but the Krebs cycle still produces electron donors. The oversupply of ETC with electron donors may increase the leaking of electrons [6], a model that has also been proposed as a mechanism of ROS overproduction in the case of hyperglycemia [8]. Besides the accumulation of electron donors, the I-R-induced oxidation of cardiolipin results in the dissociation of the ECT complex I and III from the mitochondria supercomplexes, a fact that facilitates the leak of electrons and the superoxide formation [6,9].

I-R injury results in AKI by inducing the death of renal tubular epithelial cells. Among the various types of cell death due to I-R injury, in human renal tubular epithelial cells, apoptosis and lipid peroxidation-induced cell death, known as ferroptosis, prevail [10]. AKI worsens patients’ prognosis by increasing mortality [11]. Complete recovery of kidney function requires dedifferentiation and proliferation of the remaining renal tubular epithelial cells. However, recovery is not always the case, and in addition, even after kidney function recovery, the possibility of progression to chronic kidney disease remains high [12]. The latter has been attributed to renal tubular epithelial cell senescence. I-R injury causes renal tubular epithelial cell senescence, likely through the ROS-induced DNA damage response [13,14,15,16]. Senescent cells enter a state of permanent cell cycle arrest, and in parallel, they secrete various proinflammatory and profibrotic cytokines. The first prevents recovery of renal function, while the second facilitates progression to chronic kidney disease [13,14,15,16]. Fibrosis is the hallmark of chronic kidney disease. In a murine model of unilateral renal I-R injury, senolytics reduced renal fibrosis by inducing apoptosis of senescent tubular epithelial cells [17].

In this study, we evaluated whether deceleration of the Krebs cycle reduces ROS overproduction and the subsequent cell death or senescence in cultured primary human renal proximal tubular epithelial cells (RPTECs) subjected to anoxia and reoxygenation. For this purpose, the 3-(aryloxyacetylamino)-4-hydroxybenzoic acid derivative LW6 was used to inhibit the Krebs cycle at the level of malate dehydrogenase-2 (MDH-2). LW6 was initially considered a hypoxia-inducible factor-1α (HIF-1α) inhibitor. However, later it was discovered that LW6 is a specific MDH-2 inhibitor which, by decreasing the production of ETC electron donors by the Krebs cycle, decreases oxygen consumption. The latter increases intracellular oxygen concentration resulting in HIF-1α degradation [18].

Interestingly, inhibition of MDH-2 may preserve some Krebs cycle activity preventing a total cell energy collapse [19]. This might be achieved by the pyruvate-malate cycle. In this cycle, the accumulated by the MDH-2 inhibition malate is transferred from the mitochondria to the cytosol, where it is converted by the malic enzyme to pyruvate. In its turn, pyruvate can be converted to acetyl-CoA by pyruvate dehydrogenase (PDH) and re-enter the Krebs cycle [20]. The malate/aspartate shuttle is another pathway, which involves more steps than the pyruvate-malate cycle, that can circumvent the Krebs cycle blockage at the level of MDH-2 [21]. 

Since I-R injury is implicated in the pathogenesis of AKI and many other human pathologies, researching mechanisms that may prevent I-R-induced cell death or senescence is essential and may lead to new therapeutic strategies. 

## 2. Materials and Methods

### 2.1. Cell Culture Conditions and Imaging

Primary human RPTECs (cat. no. 4100, ScienCell, Carlsbad, CA, USA) were cultured in a Complete Epithelial Cell Medium/w kit (cat. no. M6621, Cell Biologics, Chicago, IL, USA), supplemented with epithelial cell growth supplement, antibiotics, and fetal bovine serum. Cells were expanded in 75 cm^2^ flasks, and passage two cells were used for the experiments.

Cells were cultured in 6-well plates (300,000 cells per well) or 96-well plates (10,000 cells per well) for 16 h before the onset of anoxic conditions. The anoxic condition was applied for 2 h. For this purpose, the GasPak^TM^ EZ Anaerobe Container System with Indicator (cat. no. 26001, BD Biosciences, S. Plainfield, NJ, USA), which reduces oxygen levels to less than 1%, was used. Cells within the anaerobe container were cultured at 37 °C; these anoxic conditions imitate ischemia. Cells were washed, supplemented with a fresh culture medium, and placed at 37 °C in a humidified atmosphere containing 5% CO_2_; these reoxygenation conditions imitate reperfusion. 

Whenever needed, the MDH-2 inhibitor LW6 (Santa Cruz Biotechnology, Dallas, TX, USA) at a concentration of 30 μM was added at the start of the anoxic conditions and once again at the beginning of reoxygenation along with the fresh medium. Whenever indicated, 100 μM of the ferroptosis inhibitor α-tocopherol (Sigma-Aldrich; Merck Millipore, Darmstadt, Germany) was used.

Cell photos were captured with an inverted microscope (Axiovert 40C, Carl Zeiss Light Microscopy, Göttingen, Germany) and a digital camera with the related software (3MP USB2.0 Microscope Digital Camera, Amscope, Irvine, CA, USA); these experiments were repeated three times. Since live cells are necessary for reliable experimental results, all the subsequent experiments were performed at half the time required for severe RPTECs deterioration due to reoxygenation.

### 2.2. Assessment of the Proteins of Interest

The T-PER tissue protein extraction reagent (Thermo Fisher Scientific Inc., Waltham, MA, USA) supplemented with protease and phosphatase inhibitors (Sigma-Aldrich; Merck Millipore, and Roche Diagnostics, Indianapolis, IN, USA, respectively) was used for RPTECs lysis. Protein concentration was assessed with Bradford assay (Sigma-Aldrich; Merck Millipore), and 10 μg of protein from each sample were electrophoresed in sodium dodecyl sulfate-polyacrylamide gel (4–12% Bis-Tris gels, Thermo Fisher Scientific Inc.) and transferred on a polyvinylidene fluoride membrane (Thermo Fisher Scientific Inc.). The LumiSensor Plus Chemiluminescent HRP Substrate kit (GenScript Corporation, Piscataway, NJ, USA) was used for detecting the western blot bands. Whenever reprobing of the PVDF membranes was required, the Restore Western Blot Stripping Buffer (Thermo Fisher Scientific Inc.) was used. The Image J software version 1.53f (National Institute of Health, Bethesda, MD, USA) was used for analyzing the bands; these experiments were performed after 4 h of reoxygenation and repeated three times.

Each primary antibody was applied for 16 h at 4 °C, while the appropriate secondary antibody was for 30 min at room temperature. Primary antibodies were specific for the following proteins: HIF-1α (1:500, cat. no. 3716 Cell Signaling Technology, Danvers, MA, USA), 4-Hydroxynonenal (4-HNE, 1:500, cat. no. ab46545 Abcam, Cambridge, UK), p53 (1:1000, cat. no 2524, Cell Signaling Technology), activated cleaved caspase-3 (CC-3, 1:1000, cat. no. ab13847, Abcam), p21 Waf1/Cip1 (p21, 1:1000, cat. no 37543, Cell Signaling Technology), Ki-67 (Ki-67, 1:1000, cat no. NBP2-22112, Novus Biologicals, Abingdon, Oxon, UK), beta-galactosidase (GLB-1, 1:500, ab55176, Abcam), glucose transporter-1 (GLUT-1, 1:200, cat. no. sc-7903, Santa Cruz Biotechnology), hexokinase II (HK-II, 1:1000, cat. no.2867, Cell Signaling Technology), pyruvate dehydrogenase (PDH, 1:1000, cat. no. 2784, Cell Signaling Technology), PDH phosphorylated at serine 393 (p-PDH, 1:100, cat. no., orb6670, Biorbyt, Ltd., Cambridge, UK), lactate dehydrogenase-A (LDH-A, a:1000, cat. no.2012, Cell Signaling Technology), glutaminase-1 (GLS-1, 1:100, cat. no. AP18036PU-N, Acris Antibodies GmbH, Herford, Germany), glutaminase-2 (GLS2, 1:100, cat. no. AP17426PU-N, Acris Antibodies GmbH), and β-actin (1:2500, cat. no. 4967, Cell Signaling Technology). As secondary antibodies, the anti-rabbit IgG, HRP-linked antibody (1:1000, cat. no 7074, Cell Signaling Technology) or the anti-mouse IgG, HRP-linked antibody (1:1000, cat. no 7076, Cell Signaling Technology), were used.

### 2.3. Assessment of MDH-2 Activity, Cell Death, ROS, MDA, IL-1β, NADH, and ATP

The effect of LW6 on MDH-2 activity was measured in cell extracts of RPTECs cultured in 6-well plates under normoxic conditions and in the presence or not of LW6 at a concentration of 15, 30, or 60 μM. For this purpose, the mitochondrial malate dehydrogenase (MDH2) assay kit was used (cat. no. ab119693, Abcam). Before MDH-2 activity measurements, a Bradford assay was performed to adjust the lysate volume of each sample to an equal protein concentration; these experiments were repeated three times.

Besides cell imaging, cell death was also evaluated biochemically. Cells were cultured in 96-well plates and, whenever needed besides LW6, α-tocopherol was added. Cell death was assessed with the LDH release assay using the Cytotox Non-Radioactive Cytotoxic Assay kit (cat. no. G1780, Promega Corporation, Madison, WI, USA). The following equation was used Cell death (%) = (LDH in the supernatant: Total LDH) × 100; these experiments were performed after 4 h of reoxygenation and repeated three times.

ROS production was assessed in RPTECs cultured in 96-well plates. The fluorogenic probe CellROX^®^ Deep Red Reagent (cat. no. C10422, Invitrogen, Life Technologies, Carlsbad, CA, USA) at a concentration of 5 μM was added for 30 min at 37 °C. Then RPTECs were washed with PBS, and the fluorescence signal was measured on an EnSpire Multimode Plate Reader (Perkin Elmer, Waltham, MA, USA); these experiments were performed after 4 h of reoxygenation and repeated three times.

Malondialdehyde (MDA) was measured fluorometrically in cell extracts from RPTECs cultured in 6-well-plates. The Lipid Peroxidation (MDA) Assay Kit (cat. no. ab118970, Abcam) was used for this purpose. Through fluorometry, the lower detection limit of this kit is 0.1 nmol/well. Before MDA measurement, a Bradford assay was performed to adjust the lysate volume of each sample to an equal protein concentration; these experiments were performed after 4 h of reoxygenation and repeated three times.

Interleukin-1β (IL-1β) production was measured in the supernatants of RPTECs cultures in 6-well-plates. For this purpose, the commercially available Hunan IL-1β PLATINUM ELISA kit (ca. no. BMS224/2, BenderMedSystems, Vienna, Austria) was used; its sensitivity is 0.3 pg/mL; these experiments were performed after 4 h of reoxygenation and repeated three times.

Cellular NAD+, NADH, and their ratio were assessed colorimetrically in lysates from RPTECs cultured in 6-well-plates. Especially for these measurements, RPTECs were subjected only to the 2 h of anoxia and not to reoxygenation. The ChecKine NAD/NADH Assay kit was used (cat. no. KTB1020, Abbkine Scientific, Wuhan, China). The sensitivity of this kit is 0.78 nmol/mL. Before NAD+ and NADH measurements, a Bradford assay was performed to adjust the lysate volume of each sample to an equal protein concentration; these experiments were repeated three times.

Finally, cellular ATP content was assessed in RPTECs cultured in 96-well-plates using the ATP Colorimetric/Fluorometric Assay kit (cat. no. MAK190, Sigma-Aldrich; Merck Millipore). Following the manufacturer’s instructions and through fluorometry, the lower detection limit of this kit is 0.2 ng/μL; these experiments were performed after 4 h of reoxygenation and repeated three times.

### 2.4. Statistical Analysis

Data were analyzed with the IBM SPSS Statistics for Windows, Version 20 (IBM Corp., Armonk, NY, USA). The one-way analysis of variance was used for the comparison of means. For the cell imaging results, the Kruskal-Wallis H test was used. Results were presented as mean ± SEM, and a *p* < 0.05 was considered statistically significant. 

## 3. Results

### 3.1. LW-6 Inhibits MDH-2 Activity

LW6 inhibited MDH-2 activity. At the concentration of 15 μM, LW6 decreased MDH-2 activity to 69.6 ± 2.1% of the control. The concentration of 30 μM LW6 decreased MDH-2 activity further to 23.5 ± 1.3% of the control. At the highest tested concentration of 60 μM, LW6 reduced MDH-2 activity significantly to 20.9 ± 0.2% of the control, but no more than the concentration of 30 μM (Figure 1). In all other experiments, the concentration of 30 μM was used.

### 3.2. Inhibition of MDH-2 Protects RPTECs from Anoxia-Reoxygenation Injury

RPTECs cultured under normoxic conditions proliferated, reached 100% confluency at 20 h, and deteriorated at 26 h. Interestingly, the MDH-2 inhibitor LW6 did not affect the survival of RPTECs cultured under normoxic conditions, indicating the lack of toxicity of this compound at the used concentration of 30 μM. Cell imaging reveals that RPTECs previously subjected to 2 h of anoxia remained alive after 4 h of reoxygenation but deteriorated completely at 8 h (*p* < 0.05 compared to control cells). LW6 protects the subjected to anoxia-reoxygenation RPTECs, since, in this case, cells remained alive after 18 h of reoxygenation and deteriorated at 24 h (*p* < 0.05 compared to untreated subjected to anoxia-reoxygenation RPTECs) (Figure 2).

### 3.3. Anoxia-Reoxygenation Increases HIF-1α, Oxidative Stress, and p53, While LW6 Ameliorates All the Above

After 4 h of reoxygenation, the HIF-1α level increased significantly compared to its level in control RPTECs. The MDH-2 inhibitor LW6 decreased HIF-1α in RPTECs cultured under normoxic conditions and decreased it to a greater extent in RPTECs subjected to anoxia-reoxygenation (Figure 3A,B).

Reoxygenation increased ROS levels significantly, whereas LW6 ameliorated the reoxygenation-induced ROS overproduction (Figure 3C). 

Reoxygenation-induced ROS overproduction was accompanied by enhanced lipid peroxidation, as in this case, the levels of MDA, which is the end-product of the lipid peroxidation [22], increased significantly. LW6 ameliorated the reoxygenation-induced MDA increase (Figure 3D).

Similar to the MDA results, the 4-HNE modified proteins, which are aldehydic products of the lipid peroxidation [22], increased after reoxygenation, whereas the MDH-2 inhibitor LW6 ameliorated the reoxygenation-induced increase in 4-HNE modified proteins (Figure 3A,E).

Finally, the reoxygenation-induced oxidative stress was accompanied by a significant increase in p53 level, while the MDH-2 inhibitor prevented the reoxygenation-induced p53 increase (Figure 3A,F).

### 3.4. LW6 Prevents Anoxia-Reoxygenation-Induced Apoptotic or Ferroptotic Cell Death

Reoxygenation induced RPTECs apoptotic cell death as it was assessed by the levels of activated CC-3 in which all the apoptotic pathways converge [23]. The MDH-2 inhibitor LW6 prevented the reoxygenation-induced increase in CC3 levels (Figure 4A,B).

LDH release assay, which detects both necrotic and apoptotic cell death [24], showed that reoxygenation induces RPTEC death. LW6 decreased the reoxygenation-induced cell death. The ferroptosis inhibitor α-tocopherol also decreased reoxygenation-induced cell death, indicating that ferroptosis ensues during reoxygenation-induced cell injury. Interestingly, in RPTECs cultured under normoxic conditions, neither LW6 nor α-tocopherol caused cell death, indicating the lack of toxicity of these compounds at the used concentrations (Figure 4C).

### 3.5. Anoxia-Reoxygenation Induces Senescence Phenotype, While LW6 Prevents It

After 4 h of reoxygenation, the level of the cell-cycle inhibitor, p21, increased significantly. The MDH-2 inhibitor LW6 prevented the reoxygenation-induced p21 increase (Figure 5A,B).

Accordingly, to the p21 results, reoxygenation decreased the levels of Ki-67, which is a marker of cell proliferation [25]. LW6 prevented the above alteration in Ki-67 levels (Figure 5A,C).

Regarding the marker of cellular senescence GLB-1 [26], in RPTECs subjected to reoxygenation, GLB-1 increased, whereas LW6 prevented the reoxygenation-induced increase of GLB-1 (Figure 5A,D).

Besides the increase of GLB-1, reoxygenation induced in RPTECs a senescence-associated secretory phenotype [26]. Reoxygenation increased the production of the proinflammatory cytokine IL-1β. Again, the MDH-2 inhibitor LW6 significantly decreased the reoxygenation-induced overproduction of IL-1β by the RPTECs (Figure 5E).

### 3.6. Anoxia Increases NADH to NAD+ Ratio, but LW6 Decreases It

Under anoxia, cellular NADH increased, whereas NAD+ decreased, resulting in a high NADH to NAD+ ratio. The MDH-2 inhibitor LW6 prevented anoxia-induced upregulation of the NADH to NAD+ ratio (Figure 6A–C).

### 3.7. LW6 Reverses Anoxia-Reoxygenation-Induced Changes in ATP Production and Levels of Critical Enzymes of Cell Metabolism

Reoxygenation increased the RPTEC ATP significantly, while the MDH-2 inhibitor LW-6 ameliorated the above increase in cellular ATP significantly (Figure 7A).

In RPTECs subjected to reoxygenation, significant changes in the level of critical cell metabolism enzymes were observed. More precisely, reoxygenation increased the levels GLUT-1 (Figure 7B,C), HK-II (Figure 7B,D), LDH-A (Figure 7B,E), and the p-PDH to PDH ratio (Figure 7B,F). Reoxygenation did not affect the level of GLS-1 (Figure 7B,G) but increased GLS-2 significantly (Figure 7B,H). LW-6 prevented the above reoxygenation-induced changes (Figure 7B–F,H).

## 4. Discussion

I-R injury is involved in the pathogenesis of many human pathologies and is the most common cause of AKI [1,2,3,4]. Since mitochondrial ROS production during the reperfusion phase plays a significant role in cell injury [6,7,9], we evaluated in RPTECs culture whether inhibition of the Krebs cycle through downregulation of the electron donors production for the ETC ameliorates reoxygenation-induced oxidative stress and protects the cells.

For this purpose, we chose to inhibit the Krebs cycle at the MDH-2 level. Inhibition of the Krebs cycle at this point would decelerate it, but because of the pyruvate-malate cycle and the malate shuttle [20,21], the complete blockage of the Krebs cycle could be avoided saving the cells from energy collapse. LW6 is a specific MDH-2 inhibitor [18], which has been proved safe in murine experimental models [27]. For our experiments, we used LW6 at a concentration that was not toxic and inhibited MDH-2 activity at 23% of the control. In addition, by decreasing the production of electron donors for the ETC, LW6 reduces the consumption of oxygen, enhancing the intracellular oxygen concentration and leading to HIF-1α degradation [18]. 

Indeed, our experiments detected that LW6 downregulates anoxia-reoxygenation-induced HIF-1α and ROS overproduction. The decreased ROS production was accompanied by a reduction of the lipid peroxidation markers MDA and 4-HNE modified proteins. It is known that ROS overproduction causes DNA damage which in turn results in increased p53 levels [28,29]. In our experiments, reoxygenation elevated p53 levels, whereas the decrease in ROS production in RPTCs treated with LW6 prevented the reoxygenation-induced p53 increase. 

Tubular epithelial cell death is the main pathologic feature of I-R-induced AKI [4]. Cell imaging revealed that LW6 protected the cells from anoxia-reoxygenation-induced cell death. The cell imaging results were also confirmed biochemically with the LDH release assay. In the latter assay, the lipid peroxidation and ferroptosis inhibitor α-tocopherol demonstrated that ferroptotic cell necrosis contributes to RPTEC cell death. Besides ferroptosis, another kind of cell death that ensues in human RPTECs subjected to anoxia-reoxygenation is apoptosis. Accordingly, we detected that in RPTECs, anoxia-reoxygenation increases the level of activated CC3, in which all the apoptotic pathways converge [23]. LW6 prevented the reoxygenation-induced CC3 upregulation. Likely, the cause of the reoxygenation-induced apoptosis is the upregulation of p53 because of the DNA-damage response after the ROS overproduction [28,29,30]. Interestingly, the cell imaging, the LDH release assay, and the CC3 expression results in RPTECs cultured under normoxic conditions and treated with LW6 confirmed that at the used concentration LW6 is not toxic for the cells. 

Besides apoptosis, p53 can induce cell cycle arrest by upregulating p21 [28]. Indeed, we detected that anoxia-reoxygenation increases p21, whereas LW6 prevents p21 upregulation. In parallel, the cell proliferation marker Ki-67 was markedly decreased in RPTECs subjected to anoxia-reoxygenation but restored in the case of treatment with LW6. It is known that I-R-injury can induce cell senescence, a permanent cell cycle arrest state that may prevent recovery from AKI [13,14,15,16,29]. In our experiments, anoxia-reoxygenation increased the marker of cell senescence GLB-1, while LW6 prevented GLB-1 upregulation. Senescent cells acquire a senescent-associated secretory phenotype and secrete various proinflammatory and profibrotic cytokines contributing to the progression of AKI to chronic kidney disease [13,14,15,16,29]. We detected that in RPTECs subjected to anoxia-reoxygenation IL-1β production increases, while LW6 reduced IL-1β overproduction. Thus, besides protecting the RPTECs from anoxia-reoxygenation-induced cell death, inhibition of the Krebs cycle at the MDH-2 level also prevents cell senescence.

Next, we evaluated the effect of LW6 in RPTECs subjected to anoxia-reoxygenation on specific metabolic parameters. We detected that cellular NADH content increases at the end of anoxia while NAD+ decreases, likely due to the decreased NADH oxidation by the ETC [6,31]. LW6 prevented anoxia-induced upregulation of the NADH to NAD+ ratio, likely by decelerating the NAD+ reduction by the Krebs cycle reactions [6,31]. The latter is in accordance with other studies showing that LW6 and other chemically related MDH-2 inhibitors decrease mitochondrial NADH content [32]. Under reoxygenation, we found that the ATP production increases, possibly due to the accumulated electron donors (NADH and FADH_2_) during the previous anoxic phase when the Krebs cycle functioned, but the lack of oxygen inhibited ETC [6,31]. LW6 decreased cellular ATP, likely as a consequence of Krebs cycle inhibition. However, ATP remained higher than the control. This may be attributed to the fact that MDH-2 inhibition decelerates but does not stop the Krebs cycle. The latter prevents cell energy collapse and may explain the lack of LW6 cytotoxicity detected in our experiments and previous studies.

Regarding the evaluated critical enzymes of cell metabolism, anoxia-reoxygenation increased the levels of GLUT-1, HK-II, LDH-A, the ratio of p-PDH to PDH, and the level of GLS-2. The increased HIF-1α may explain the rise of GLUT-1, HK-II, and LDH-A since HIF-1α transcribes the related genes [33]. Also, the anoxia-reoxygenation-induced HIF-1α increase may explain the increase of the p-PDH to PDH ratio since it upregulates PDH kinase, which phosphorylates and inhibits PDH [33,34]. The increased GLS-2 could be attributed to p53 since the latter transcribed its gene [35]. LW6 restored all the changes mentioned above, likely by its effect on HIF-1α and p53 levels. It seems that under reoxygenation, more glucose enters the RPTECs and is preferentially catabolized through the energetically less efficient but faster anaerobic glycolysis [31,33]. However, anaplerosis of the Krebs cycle increased since GLS-2, the first enzyme in the glutaminolysis anaplerotic pathway, rises [31,35]. LW6 decreases glucose entrance into RPTECs, and glutaminolysis. On the other hand, by decreasing the p-PDH to PDH ratio, LW6 accelerates pyruvate transformation to acetyl-CoA [31,33,34]. The latter facilitates glucose catabolism through the energetically more efficient Krebs cycle and the pyruvate-malate cycle as well. By doing so, the MDH-2 inhibitor LW6 prevents total energy cell collapse. Figure 8 depicts in an abstract form the evaluated pathways.

A limitation of our study lies within its in vitro nature. A recent transcriptomic comparison detected differences between fourteen available proximal tubular epithelial cell culture models from six species and the intact kidney [36]. Although we used primary cells and not immortalized cell lines, this remains a limitation. However, the strict conditions of the cell culture systems can exclude many confounding factors and allow a more accurate evaluation of specific molecular pathways. Thus, our study could be considered as a starting point for additional in vivo studies that will evaluate the possible benefits of hacking cell metabolism at the MDH-2 level to ameliorate I-R injury.

In conclusion, in RPTECs, inhibition of the Krebs cycle at the level of MDH-2 decreases oxidative stress without causing cell energy collapse and prevents anoxia-reoxygenation-induced cell death or senescence. Thus, MDH-2 may be a promising pharmaceutical target against I-R-induced AKI.

## Figures and Tables

**Figure 1 biomolecules-12-01415-f001:**
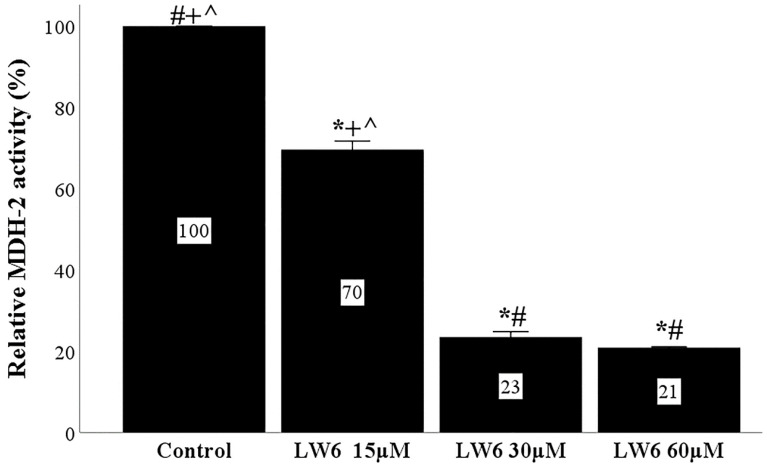
LW6 inhibits MDH-2 activity. MDH-2 activity was assessed in cell lysates from RPTECs cultured under normoxia with or without LW6 at a concentration of 15, 30, or 60 μM. LW6 inhibited MDH-2 activity in all concentrations, with the highest inhibition achieved at 30 and 60 μM. * *p* < 0.05 vs. control; # *p* < 0.05 vs. RPTECs treated with 15 μM LW6; + *p* < 0.05 vs. RPTECs treated with 30 μM LW6; ^ *p* < 0.05 vs. RPTECs treated with 60 μM LW6. MDH-2, malate dehydrogenase 2.

**Figure 2 biomolecules-12-01415-f002:**
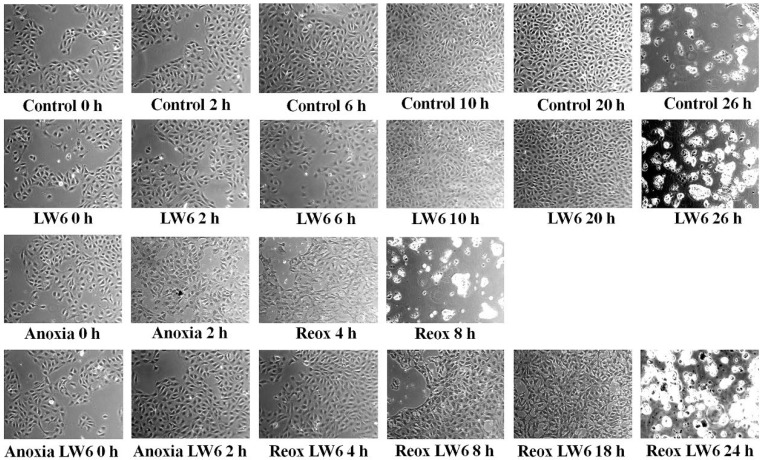
LW6 protects RPTECs from anoxia-reoxygenation cell death. Cell imaging (magnification ×100) revealed that anoxia-reoxygenation induces RPTEC death, whereas the MDH-2 inhibitor LW6 prevents anoxia-reoxygenation-induced cell death.

**Figure 3 biomolecules-12-01415-f003:**
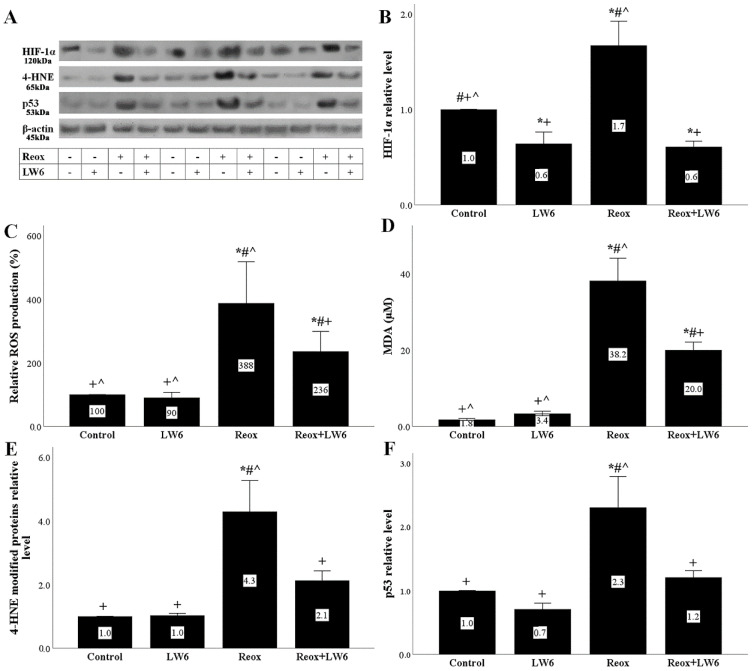
Anoxia-reoxygenation increases HIF-1α, oxidative stress, and p53, while LW6 alleviates all the above. Anoxia-reoxygenation increased HIF-1α, but the MDH-2 inhibitor decreased HIF-1α (**A**,**B**). Reoxygenation increased ROS production, while LW6 ameliorated reoxygenation-induced ROS overproduction (**C**). Reoxygenation enhanced cellular MDA level, while LW6 ameliorated reoxygenation-induced MDA upregulation (**D**). Reoxygenation increased the level of 4-HNE modified proteins, whereas LW6 reduced the level of reoxygenation-induced 4-HNE modified proteins (**A**,**E**). Reoxygenation upregulated the p53 level, whereas LW6 prevented its upregulation (**A**,**F**). * *p* < 0.05 vs. control; # *p* < 0.05 vs. LW6-treated RPTECs under normoxia; + *p* < 0.05 vs. RPTECs under reoxygenation, ^ *p* < 0.05 vs. LW6-treated RPTECs under reoxygenation. HIF-1α, hypoxia-inducible factor-1α; ROS, reactive oxygen species; MDA, malondialdehyde; 4-HNE; 4-hydroxynonenal.

**Figure 4 biomolecules-12-01415-f004:**
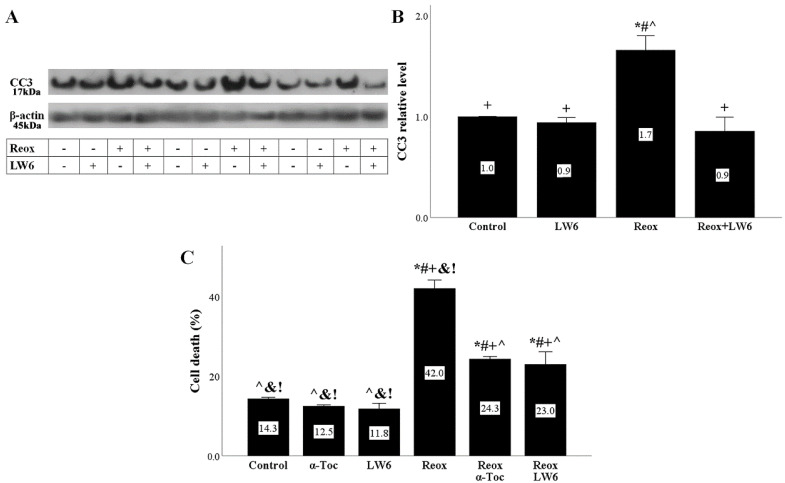
LW6 prevents anoxia-reoxygenation-induced apoptotic or ferroptotic cell death. Anoxia-reoxygenation increased CC3 level, whereas the MDH-2 inhibitor LW6 prevented the upregulation of this pro-apoptotic caspase (**A**,**B**). * *p* < 0.05 vs. control; # *p* < 0.05 vs. LW6-treated RPTECs under normoxia; + *p* < 0.05 vs. RPTECs under reoxygenation, ^ *p* < 0.05 vs. LW6-treated RPTECs under reoxygenation. Anoxia-reoxygenation induced cell death, which was ameliorated by the LW6 or the ferroptosis inhibitor α-tocopherol (**C**). * *p* < 0.05 vs. control; # *p* < 0.05 vs. α-Toc-treated RPTECs under normoxia; + *p* < 0.05 vs. LW6-treated RPTECs under normoxia, ^ *p* < 0.05 vs. RPTECs under reoxygenation; & *p* < 0.05 vs. α-Toc-treated RPTECs under reoxygenation; ! *p* < 0.05 vs. LW6-treated RPTECs under reoxygenation. CC3, activated cleaved caspase-3; α-Toc, α-tocopherol.

**Figure 5 biomolecules-12-01415-f005:**
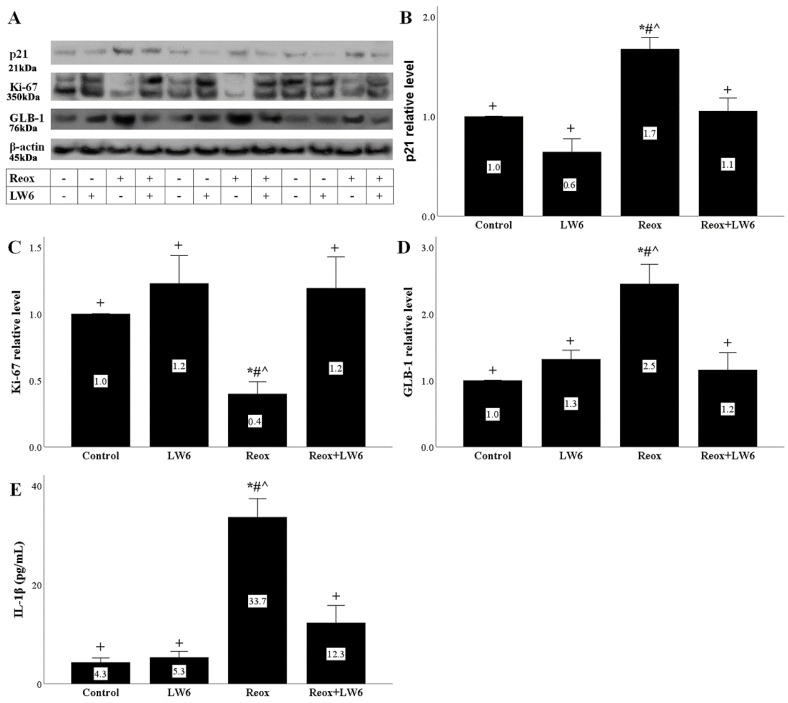
Anoxia-reoxygenation induces senescence phenotype, while LW6 prevents it. Anoxia-reoxygenation increases the p21 level, whereas the MDH-2 inhibitor LW6 prevents reoxygenation-induced p21 upregulation (**A**,**B**). Reoxygenation downregulated Ki-67, while LW6 prevented reoxygenation-induced Ki-67 reduction (**A**,**C**). Under reoxygenation, GLB-1 increased, but LW6 reversed the above change (**A**,**D**). RPTECs under reoxygenation produced more IL-1β, whereas LW6 downregulated reoxygenation-induced IL-1β overproduction (**E**). * *p* < 0.05 vs. control; # *p* < 0.05 vs. LW6-treated RPTECs under normoxia; + *p* < 0.05 vs. RPTECs under reoxygenation, ^ *p* < 0.05 vs. LW6-treated RPTECs under reoxygenation. GLB-1, beta-galactosidase; IL-1β, interleukin-1β.

**Figure 6 biomolecules-12-01415-f006:**
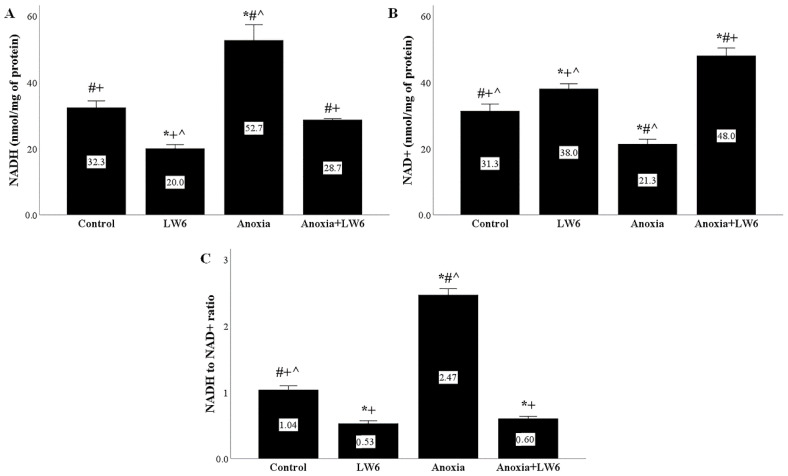
LW6 prevents anoxia-induced upregulation of NADH to NAD+ ratio. In RPTECs subjected to anoxia, cellular NADH content increased (**A**), NAD+ decreased (**B**), and the NADH to NAD+ ratio became high (**C**). The MDH-2 inhibitor LW6 prevented the anoxia-induced upregulation of NADH to NAD+ ratio (**C**). * *p* < 0.05 vs. control; # *p* < 0.05 vs. LW6-treated RPTECs under normoxia; + *p* < 0.05 vs. RPTECs under anoxia, ^ *p* < 0.05 vs. LW6-treated RPTECs under anoxia. NADH, reduced nicotinamide adenine dinucleotide; NAD+ oxidized nicotinamide adenine dinucleotide.

**Figure 7 biomolecules-12-01415-f007:**
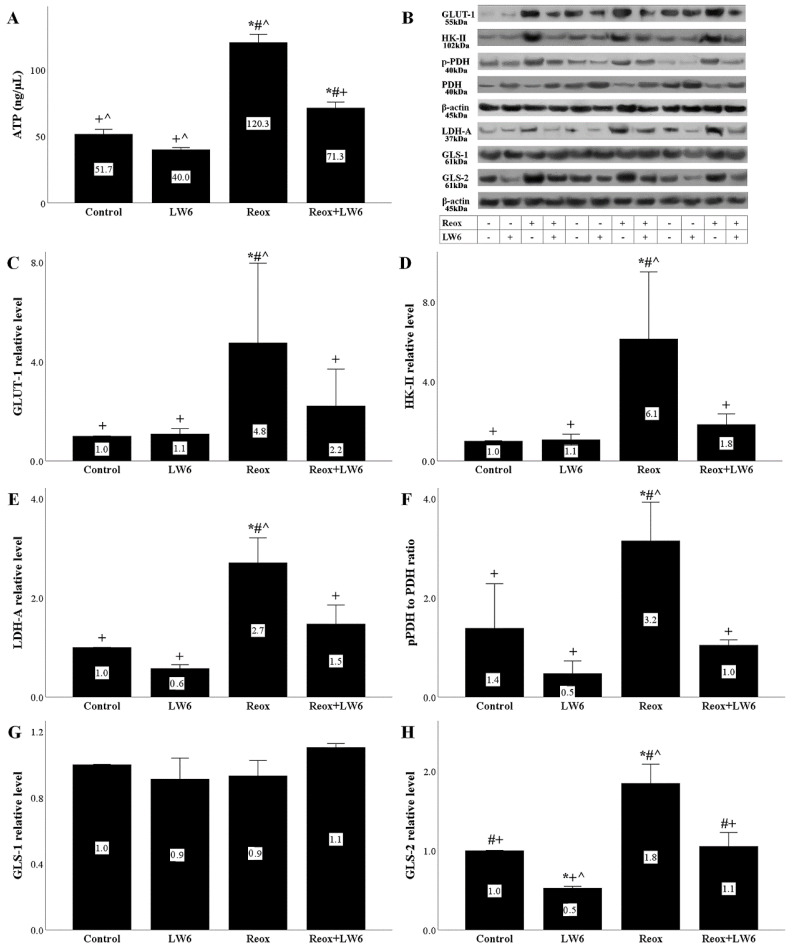
LW6 reverses anoxia-reoxygenation-induced changes in ATP production and levels of critical enzymes of cell metabolism. Under reoxygenation, cellular ATP content increases in RPTECs. The MDH-2 inhibitor LW6 reduced the reoxygenation-induced increase in ATP (**A**). Anoxia-reoxygenation increased GLUT-1, whereas LW6 reversed this change (**B**,**C**). Similar results were obtained for HK-II (**B**,**D**), LDH-A (**B**,**E**), p-PDH to PDH ratio (**B**,**F**), and GLS-2 (**B**,**H**). GLS-2 level remained unaffected (**B**,**G**). * *p* < 0.05 vs. control; # *p* < 0.05 vs. LW6-treated RPTECs under normoxia; + *p* < 0.05 vs. RPTECs under reoxygenation, ^ *p* < 0.05 vs. LW6-treated RPTECs under reoxygenation. GLUT-1, glucose transporter-1; HK-II, hexokinase II; LDH-A, lactate dehydrogenase-A; PDH, pyruvate dehydrogenase, p-PDH, PDH phosphorylated at serine 393; GLS-1, glutaminase-1; GLS-2, glutaminase-2.

**Figure 8 biomolecules-12-01415-f008:**
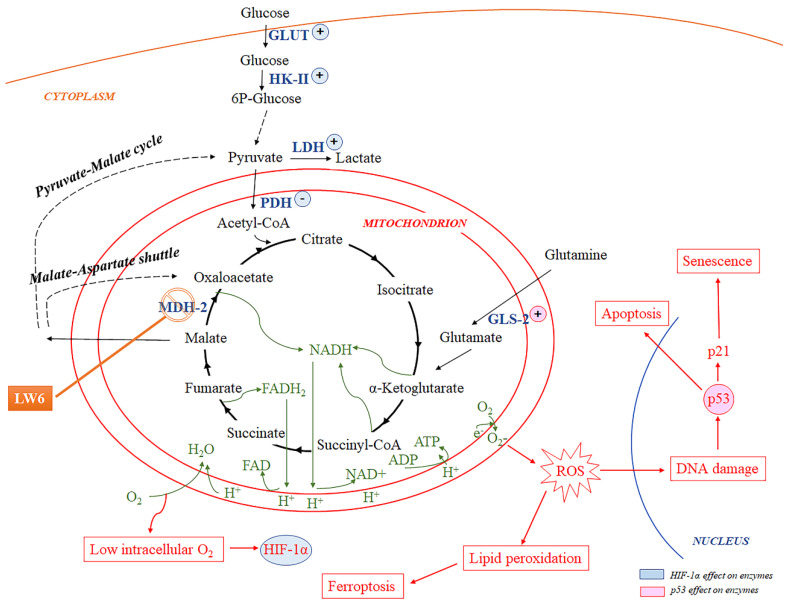
Inhibition of MDH-2 affects cell metabolism and fate. The accumulated during anoxia electron donors result in ROS overproduction during the reoxygenation and eventually in cell death through apoptosis and ferroptosis or cell senescence. Although by inhibiting the Krebs cycle, LW6 decreases electron donor production in the anoxia phase and protects the cells during reoxygenation, ATP is still produced, preventing cell energy collapse. In the latter, the pyruvate-malate cycle and the malate-aspartate shuttle may contribute, as well as the effect of LW6 on the transcription factors HIF-1α and p53. Both are downregulated by LW6 and affect the expression of critical enzymes of cell metabolism. In addition, downregulation of p53 by LW6 protects the cells from apoptotic cell death or senescence, while downregulation of ROS prevents ferroptotic cell death. FADH_2_, reduced flavin adenine dinucleotide; GLS-2, glutaminase-2; GLUT, glucose transporter; HIF-1α, hypoxia-inducible factor-1α; HK-II, hexokinase-II, LDH, lactate dehydrogenase; MDH-2, malate dehydrogenase; PDH, pyruvate dehydrogenase; NADH, reduced nicotinamide adenine dinucleotide; ROS, reactive oxygen species.

## Data Availability

The analyzed datasets generated during the study are available from the corresponding author upon reasonable request.

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
