# Peer review of "Inhibition of Malate Dehydrogenase-2 Protects Renal Tubular Epithelial Cells from Anoxia-Reoxygenation-Induced Death or Senescence"

_biomolecules, 2022, doi:10.3390/biom12101415_

Round 1
Reviewer 1 Report
Eleftheriadis and coworkers have submitted a nicely performed study examining the effect of inhibition of MDH-2 on IR injury. Their rationale is clearly laid out. Their experiments are sound.
This reviewer has 3 major issues with the paper
1. This is a study of metabolism performed in a cell culture model. Recent studies have highlighted the changes in metabolism that occur when kidney tubular cells in culture are studied compared to intact kidney (Khundmiri et al, J Am Soc Nephrol, 2021). While this observation does not nullify the entire study, it is very important for the authors to acknowledge this significant limitation. The manuscript would be considerably stronger if studies in intact kidneys could be incorporated.
2. Because the authors are examining the effect of an enzyme that influences mitochondrial activity, this reviewer was surprised that the effect on mitophagy was not examined. Is it possible that inhibition of MDH-2 has a beneficial effect on the degree of mitophagy response, thus leading to some preservation of function and blocking IR-induced death?
3. This reviewer did not see where MDH-2 activity had actually been measured to confirm that the LW6 was having the desired effect.
4. Additional considerations.
a. The authors may want to consider including a figure that describes their model for how MDH-2 protects from IR injury, i.e., which pathways is it affecting, etc.
b. A dose response for the effect of LW6 would strengthen the assertion that this is a response that is dependent on the desired effect of LW6
Author Response
We would like to thank the reviewer for the valuable comments that helped us to improve our study.
The limitation of the in vitro nature of our study has been further highlighted in the revised manuscript and the suggested article has been cited. “A limitation of our study lies within its in vitro nature. A recent transcriptomic comparison detected differences between fourteen available proximal tubular epithelial cell culture models from six species and the intact kidney [36]. Although we used primary cells and not immortalized cell lines, this remains a limitation. However, the strict conditions of cell culture systems can exclude many confounding factors and allow a more accurate evaluation of specific molecular pathways. Thus, our study could be considered as a starting point for additional in vivo studies that will evaluate the possible benefits of hacking cell metabolism at the MDH-2 level to ameliorate I-R injury.”
We aimed to test the hypothesis that partial Krebs cycle inhibition by LW6 decreases the accumulation of electron donors (NADH and FADH2) during the anoxia phase, the known reductive stress, a fact that ameliorates ROS overproduction (oxidative stress) during the following reoxygenation phase and protecting the cells from I-R-injury. We did an extra experiment to support our hypothesis. Mitophagy was not involved in our original hypothesis. Instead, apoptosis, ferroptosis, and senescence were evaluated.
Besides literature data, the fact that LW6 inhibits MDH-2 is supported by our experimental results. In our study, LW6 downregulated HIF-1α, and it is known that this is due to inhibition of MDH-2, deceleration of the Krebs cycle, decreased electron donor production for the ETC, and less oxygen consumption by the oxidative phosphorylation. The increased intracellular oxygen results in HIF-1α degradation. The above, along with the related literature, have been noted in the manuscript. Also supporting the effectiveness of LW6 on MDH-2 inhibition are the results of an extra experiment we performed and included in the revised manuscript. We measured NADH/NAD+ ratio at the end of the anoxia. Indeed, we found that anoxia increases this ratio, whereas the MDH-2 inhibitor LW6 did not allow the NADH/NAD+ ratio to increase. We added these data along with the related figure (Figure 5) in the revised manuscript.
In the revised manuscript, according to the right reviewer's recommendation, we added a figure (Figure 7) that depicts the many evaluated molecular pathways.
We did not use different LW6 concentrations for all the experiments because we had performed preliminary cell imaging experiments, which showed the lack of toxicity and cell preservation after reoxygenation at the concentration of 30 μΜ. In addition, the presented in the manuscript cell imaging, LDH release assay, and CC3 results confirmed that at the used concentration LW6 was not toxic for the RPTECs. Finally, the other results showed that at the used concentration, LW6 is effective.
Reviewer 2 Report
The authors presented evidence that inhibition of MDH2 prevents renal tubular cell damage induced by anoxia-reoxygenation. While the study is purely in vitro by using cells, the data indeed support the conclusion. One concern is the electron donor accumulation discussed, but the authors never presented the data. This could be easily done by measuring NADH/NAD+ ratio. This would tell NADH is indeed higher in the authors system under the experimental conditions.
Author Response
We want to thank the reviewer for the encouraging comments and the critical suggestion. We performed an extra experiment to measure NADH/NAD+ ratio at the end of the anoxia. Indeed, we found that anoxia increases this ratio, whereas the MDH-2 inhibitor LW6 did not allow the NADH/NAD+ ratio to increase. We added these data along with the related figure (Figure 5) in the revised manuscript.
Round 2
Reviewer 1 Report
The reviewer thanks the authors for their revisions. the one issue that remains that should be addressed is the dose of LW6. it is reasonable to present a dose response for any agent used. The concern in this case would not be "is LW6 toxic?" but is it acting by some mechanism other than that expected by the authors? demonstration of inhibition of the enzyme activity and presentation of a dose response effect would strengthen the work considerably.
Author Response
Dear Reviewer,
In the revised manuscript we provide new data from an additional experiment, which directly confirms the inhibitory effect of LW6 on MDH-2 activity of RPTECs at the concentration used in our experiments. The related information is included in the methods and the results sections, depicted in a figure, and commented on in the discussion of the revised manuscript.
Thank you for the valuable comments,
Best Regards,
Theodoros Eleftheriadis